# Acute and Subacute Toxicity Evaluation of Erythrocyte Membrane-Coated Boron Nitride Nanoparticles

**DOI:** 10.3390/jfb14040181

**Published:** 2023-03-25

**Authors:** Jinfeng He, Xuanping Zhang, Linhong Liu, Yufei Wang, Renyu Liu, Min Li, Fuping Gao

**Affiliations:** 1Department of Basic Medicine, Shanxi Medical University, Taiyuan 030001, China; hejf@ihep.ac.cn (J.H.); wang_yufei0047@163.com (Y.W.); 2CAS Key Laboratory for Biomedical Effects of Nanomaterial and Nano Safety, Institute of High Energy Physics, Chinese Academy of Sciences, Beijing 100049, China; liulh@ihep.ac.cn (L.L.); limin@ihep.ac.cn (M.L.); 3Jinan Laboratory of Applied Nuclear Science, Jinan 251401, China

**Keywords:** functional nanoparticle, biocompatibility, BN@RBCM, toxicity evaluation

## Abstract

Boron nitride nanoparticles have been reported for boron drug delivery. However, its toxicity has not been systematically elucidated. It is necessary to clarify their potential toxicity profile after administration for clinical application. Here, we prepared erythrocyte membrane-coated boron nitride nanoparticles (BN@RBCM). We expect to use them for boron neutron capture therapy (BNCT) in tumors. In this study, we evaluated the acute toxicity and subacute toxicity of BN@RBCM of about 100 nm and determined the half-lethal dose (LD50) of the particles for mice. The results showed that the LD50 of BN@RBCM was 258.94 mg/kg. No remarkable pathological changes by microscopic observation were observed in the treated animals throughout the study period. These results indicate that BN@RBCM has low toxicity and good biocompatibility, which have great potential for biomedical applications.

## 1. Introduction

Hexagonal boron nitride nanoparticles (BNNPs) are a novel nanomaterial named white graphite due to the layered structure analogs of graphene, but with the complete substitution by alternating B and N atoms [1]. Due to their extraordinary properties, nanoparticles have attracted much attention [2,3,4,5]. BN colloids can reach tens of nanometers with good crystallinity and high thermal neutron capture cross section [6]. As a hot type of radiotherapy, successful boron neutron capture therapy (BNCT) depends on the high accumulation of effective and low-toxicity boron-containing drugs in the tumor [7,8]. Some studies have reported the modification of boron nitride nanoparticles to enhance their tumor accumulation for BNCT [6,9,10,11]. With more and more studies on functionalized nanomaterials have been reported, nanomaterials properties may be altered to break through lots of gaps, such as crossing the blood–brain barrier (BBB) [12], targeting [13], drug delivery, detoxification, vaccination [14], and others. We then learned that different classes of natural biocomponents such as plants, fungi, bacteria, and yeast have been employed as efficient resources for the fabrication of NPs [15], offering opportunities for the targeted delivery of various therapeutic agents. Therefore, our aim is to apply positive modification to nanomedicine. Inspired by the natural nature of biomembranes, it has been shown that the use of cell membrane coating to endow nanoparticles with natural cell membrane functions can improve the biocompatibility and pharmacokinetics of nanoparticles and enhance therapeutic efficacy [16]. In addition, polyethylene glycol (PEG) has often been coated on the surface of pristine BN to improve the phase change behavior and enhance solubility and biocompatibility [17]. These coatings would not alter the structure of BN, further improving the delivery efficiency while retaining the property and character of BN.

Therefore, in this study, we prepared the erythrocyte membrane and 1,2-distearoyl-sn-glycero-3-phosphoethanolamine-N[carboxyl (poly-(ethylene glycol)2000] (DSPE-PEG2000-COOH) to coat boron nitride nanoparticles for future application in BNCT and tumor treatment. To date, the overall toxicity of BNs in mice has remained largely unexplored, and most studies have primarily focused on in vitro cytotoxicity [18,19], so further investigation into the toxicity of BN nanoparticles (BNNPs)—especially for in vivo use—was desired. To date, various studies on the toxic effects of BN nanotubes (BNNTs) in cell lines are encouraging and once again confirm BNNTs as an innovative nanomaterial for potential biomedical applications [20,21]. BN nanospheres also showed no apparent cytotoxicity and were considered more suitable than BNNTs for biomedical applications [22]. Li et al. confirmed the biocompatibility of functionalized boron nanomaterials. However, the application of functionalized BNNPs in nanomedicine still remained largely unexplored [6,10,23]. NPs have improved tumor accumulation when enveloped by different types of cancer cell membranes (CCM), erythrocyte membranes, exosomes, and other biomimetic camouflage by self-recognition and internalization for and prolonging blood circulation time and improving tumor accumulation [24]. These modifications further helped to effectively transport anticancer drugs into tumor cells and suppress tumor cell growth.

Here, we used membrane materials derived from red blood cells (RBCs) to coat BN nanoparticles (named BN@RBCM). We monitored and recorded physical signs after a single injection for up to 14 days, while also assessing the toxic effects of multiple administrations. A thorough in vivo study of the toxicity of BN@RBCM is essential to understand its potential risk and provide useful information for its safe application. However, to the best of our knowledge, systematic in vivo toxicity evaluation of BN@RBCM hasn’t been reported. Feng et al. evaluated the toxicity of erythrocyte membrane-coated boron nitride nanospheres (CM-BNNS) [22]; however, the in vivo toxicity evaluation only focusing on a single low dose level could not fully reflect the in vivo toxicity of CM-BNNS.

In this study, Kunming mice were used to investigate toxicity by intravenous administration of BN@RBCM at various concentrations. Toxicity was assessed by mortality, body weight, food consumption, hematology and blood chemistry, and pathological examination of organs and tissues. We evaluated not only the acute toxicity of a single dose, but also the subacute toxicity of multiple doses of BN@RBCM using dose gradients at different levels and obtained the half-lethal dose (LD50) of BN@RBCM.

## 2. Experimental

### 2.1. Materials

All chemical reagents were directly used without further purification. Boric-10 acid (H_3_^10^BO_3)_ was purchased from Liaoning Honghao Chemical Co., Ltd., Liaoyang City, China, melamine was obtained from Macklin Reagent Co., Ltd., and DSPE-PEG2000-COOH was purchased from Xi’an Ruixi Biotechnology Co., Ltd., Xi’an City, China. The phosphate buffer solution (PBS) was obtained from Meilun Biotechnology Co., Ltd, Dalian, China.

### 2.2. Synthesis and Characterization of Boron Nitride Nanoparticles

Boron nitride nanoparticles (BNNPs) were synthesized by pyrolysis at 1100 °C followed by solvent cutting as reported in the literature. Briefly, boric acid and melamine were ground with a pestle and mortar in a molar ratio of 1:6 [25]. The precursor was then placed in a horizontal tube furnace (Φ 25 mm, 1200 mm in length) at a heating rate of 10 °C/min to 1100 °C and maintained under Ar flow for 2 h. The crude samples obtained were then dispersed in water. The 60 mg of the as-prepared powders, termed CBN1100, were sheared by “cutting solvents” at 80 °C for 2 h. The cutting solvents used were 30 mL of water. The resulting BNNP dispersion was then centrifuged at 1467× *g* for 10 min to remove the impurities and remaining bulk crystals, then dialyzed for 48 h (membrane cut-off: 3500 Da) and filtered (0.45 μm). The resulting samples were designated BN1100.

### 2.3. Red Blood Cell Membrane Isolation

The red blood cell (RBC) membrane-derived vesicles were prepared using hypotonic hemolysis [26], and whole blood was collected from Kunming mice in an anticoagulation tube with low molecular weight heparin solution. Plasma and buffy coat were carefully removed by centrifugation at 1123× *g* for 10 min at 4 °C. The resulting erythrocytes were washed with cold saline. Then, 0.25 × PBS in a ratio of 20:1 was added for hemolysis at 4 °C for 12 h. The solution was centrifuged three times at 12,000× *g* for 20 min to remove the released hemoglobin. Finally, the light pellet was resuspended in 1 × PBS.

### 2.4. Surface Modification of BNNPs

To form the stable DSPE-PEG2000-COOH-inserted RBC ghosts, the RBC membrane was dispersed in deionized water by sonication and then incubated with DSPE-PEG2000-COOH at 4 °C for 30 min. The freshly prepared DSPE-PEG2000-COOH-inserted RBC membrane was mixed in the specific ratio to coat the surface of the BNNPs [27]. After the mixture was again incubated overnight, the solution was purified by centrifugation and washing three times with PBS in an ultrafiltration tube. The residual products were designated as BN@RBCM for future use.

### 2.5. Experimental Animals

Kunming (KM) mice, half male and half female, weighing 20–25 g, were purchased from Huafukang Bioscience Company (Beijing, China). All animals were treated in accordance with the regulations of the National Act on the Use of Experimental Animals (China) and were approved by the Institutional Animal Care and Ethics Committee of the Chinese Academy of Sciences.

### 2.6. Acute Toxicity Experiment

Acute toxicity was studied in 54 healthy 4-week-old mice. The mice were randomly divided into 8 experimental groups (*n* = 7 per group) after overnight fasting from food but not water. The mice were administered BN@RBCM via the tail vein at doses ranging from 82.35 to 700 mg/kg. The dose was diluted at a ratio of 1:0.7 and the dose was tapered. The control group received distilled water. After one injection, clinical signs of toxicity were observed for 14 days. Body weight, food consumption, and survival state were recorded for analysis. At the end of the experiment, blood and major organs were collected for analysis of any changes. The half-lethal dose (LD50) was determined by mouse mortality. The LD50 and confidence interval (CI 95%) were calculated by Equations (1)–(4).
M = X_m_ − i (∑p − 0.5)(1)
LD_50_ = lg^−1^ M,(2)
(3)$$S_{\mathrm{lgLD}50}\;=i\sqrt{\sum (p\times q)/n}$$
CI = lg^−1^(lg LD50 ± 1.96 × S _lg LD50_)(4)

X_m_ is the log value of the maximum dose, i is the log dose difference between two adjacent dose groups (i = 0.155), p is mortality, q is survival, and n is the number of animals in each group.

### 2.7. Subacute Toxicity Experiment

As for the trial period of subacute toxicity, we adhered to the following criteria: ISO 10993-11, GB/T 16886.11, USP 88, and DIN EN ISO 10993-11-2009 of the biological evaluation of medical devices—Part 11: systemic toxicity tests. For the subacute toxicity study, a total of 40 mice, half male and half female (20–25 g), were randomly divided into 4 experimental groups (*n* = 10 per group) and administered 0 (saline), 26.93, 67.33, and 134.65 mg/kg of BN@RBCM once every other day for 14 days. The observation in mice of any physiological and behavioral changes was recorded. On the 15th day, the mice were euthanized and blood and vital organs were collected for hematological, blood biochemical, and histopathological analysis. For hematological analysis, blood was collected in an EDTA-coated tube and blood cell counts were measured using an automated blood cell analyzer (Mindray, BC-2800 vet). For biochemical analysis, whole blood samples were separated by centrifugation at 4 °C at 825× *g* rpm for 15 min. The supernatant was analyzed for blood urea nitrogen (BUN), creatinine (CRE), aspartate aminotransferase (AST), alanine aminotransferase (ALT), and total bilirubin (TBIL) using an automated biochemistry analyzer (Chemray240, Chemray800, Rayto Life and Analytical Sciences Co., Ltd., Shenzhen, China). Isolated organs and tissues were washed with PBS, weighed, and macroscopically examined for gross changes. The relative organ weight (ROW) of each animal was then calculated as follows ROW = [absolute organ weight (mg)/bodyweight (g)] × 100.

### 2.8. Histopathological Analysis

After fixation in 4% paraformaldehyde, organs/tissues were embedded in paraffin and sectioned at 5–10 μm thickness. Slides were stained with hematoxylin and eosin (H&E) and histopathologically examined by light microscopy (×200 and ×400) for any abnormalities.

### 2.9. Statistical Analysis

Statistical analysis was performed using Origin2018 software (Copyright © 2018 by OriginLab Corporations permission). Data were analyzed using the Student *t*-test. The results are presented as mean ± standard deviation of at least triplicate measurements, and *p* < 0.05 was considered a significant difference.

## 3. Results

### 3.1. Characterization of BN@RBCM

The purified BN was obtained by cutting crude BN into small nanofragments using H_2_O as a strong polar solvent and then dialysis in water to remove impurities. The morphology characteristic of BN and BN@RBCM was evaluated by transmission electron microscopy (TEM) and the average diameter was measured by dynamic light scattering (DLS). The TEM image of bare BN (Figure 1a) showed an approximately circular or hexagonal shape with an average size of about 10 nm, and synthesized BN@RBCM negatively stained with phosphortungstic acid (Figure 1b) showed a distinct membrane-like structure around the BN nanoparticles with a size of about 100 nm, probably due to some encapsulated nanosheets. Magnification of the marked RBC membrane coating on BN showed a significant core-shell structure, demonstrating the successful coating of the erythrocyte membrane (Figure 1c). The results of dynamic light scattering showed that the average size of BN@RBCM was 137.71 ± 3.36 nm, while the average size of naked BNNPs was 125.81 ± 0.86 nm. The increase in the size of BN@RBCM further confirmed the successful coating of the erythrocyte membrane (Table 1). The slight increase in particle size obtained by DLS compared with that shown by TEM is likely due to the hydrodynamic particle size detected by DLS.

### 3.2. Acute Toxicity of BN@RBCM in Mice

Different injected doses of BN@RBCM and mortality are summarized in Table 2. We recorded acute toxic reactions of a single injection, including twitching, entasia, tachypnea, absence of reflex, no eyelid reflex, and total death within 6 h. The results showed that mice started to die at a dose of 168 mg/kg and all experimental mice died at a dose of more than 490 mg/kg. The remaining mice were still given water and food. They returned to normal after a few hours to 14 days of observation. We found that the treatment groups had the same trend in weight gain and body weight changes compared with those of the control group in the remaining mice, and these mice in the treatment groups were not obviously different in food consumption per animal/day (Figure 2a–c). The survival of mice treated with different doses is shown in Figure 2d. It clearly showed that death occurred at a dose above 168.07 mg/kg. The result demonstrated only in the higher dose group did the BN@RBCM have significant toxic effects. The approximate 50% lethal dose (LD50) of BN@RBCM was 258.94 mg/kg, with a 95% confidence interval (CI) of 212.0313–316.2278 mg/kg.

At the end of the observation period, vital tissues and organs were removed from the mice. Taking the organs from mice treated with BN@RBCM at 117.65 mg/kg as a representative, there was no obvious abnormality in the appearance of the organs compared with the control (Figure 3a), and the organ coefficient in terms of organ weight to body weight had no significant difference compared with that of the control group (Figure 3b). The organ coefficients of the other groups are detailed in Table 3. There was no statistically significant difference between the organ coefficients of the surviving mice in the treatment group with a dose below 240 mg/kg and that of the control group.

As a sensitive target for toxic compounds, the hematopoietic system is an important index for determining physiological and pathological states. The results of hematological and blood biochemical analyses, represented by the 117.65 mg/kg dose group, are summarized in Table 4. The parameters, including liver function markers (aspartate aminotransferase (AST), alanine transaminase (ALT), and total bilirubin (TBIL)) and kidney function markers (creatinine (CREA) and blood urea nitrogen (BUN)), remained unchanged when compared with saline-treated mice (Figure 4a–c), there was no statistically significant difference in biochemical parameters compared with that of the control group (*p* > 0.05). Furthermore, the clinical hematological parameters shown indicated no significant difference between the different groups. The histological examination of the major organs, including the heart, liver, spleen, lung, and kidney, showed no atrophy of the glomeruli, degeneration of the renal tubules, hemorrhage, or inflammatory infiltration in the interstitial tissues in the mice injected with BN@RBCM (117.65 mg/kg). The liver sections of the mice injected with BN@RBCM (117.65 mg/kg) showed a normal hepatic lobule architecture. The central vein in the lobule is surrounded by hepatocytes and with strongly eosinophilic granulated cytoplasm as well as distinct nuclei [28]. These results indicated no morphological or microstructural changes compared with those of the control (Figure 4d). We also detected hematology, blood biochemistry, and histopathology of the remaining surviving mice in the 168.07 mg/kg group and 240.1 mg/kg dose groups, respectively, and no abnormalities were found. The death rate of the mice in the group of 343 mg/kg, 490 mg/kg, and 700 mg/kg was too high for further histopathological tests.

### 3.3. Subacute Toxicity of BN@RBCM in Mice

Next, we investigated the subacute toxicity of BN@RBCM in mice by repeated administration. The mice were administered 25.89, 64.74, and 129.47 mg/kg of BN@RBCM, respectively, once every other day for 14 days. All animals survived in this study and no signs of toxicity were observed during the two weeks of treatment. In assessing drug-induced toxicity, changes in general behavior and body weight were considered to be early signs. The effect of different doses of BN@RBCM on body-weight gain was illustrated in Figure 5a. Notably, the mean body weights of the mice were not adversely affected after the administration of BN@RBCM for 14 days. It is noteworthy that treated animals did not differ from the control animals in terms of changes in relative organ weights (Figure 5b and Table 5). The BN@RBCM nanomaterials showed no subacute toxic abnormal clinical signs.

As a further evaluation of in vivo toxicity, we examined the hematological and blood biochemical markers of mice after 14 days of repeated administration of BN@RBCM. Although the HCT and MCHC value of the mice treated with BN@RBCM was out of the normal range, there was no significant difference compared with the control group. The other hematological indices of the BN@RBCM-treated mice were within the normal range (Table 6). Similarly, the levels of white blood cells, lymphocytes, and platelets in mice at different doses did not show significant abnormalities compared with controls. The results indicated that BN@RBCM had no significant hematological toxicity. The analyses of blood biochemical indices of the treatment groups showed that AST and ALP, as well as the levels of total serum bilirubin, BUN, and CREA had no obvious abnormality compared with that of the control group, and all the values were in the normal range, which provided powerful evidence indicating no toxic effects of BN@RBCM on liver and kidney function (Figure 6). Histopathological examination of the liver and kidney did not reveal inflammatory infiltration in the interstitial tissues, edema, or necrotic histocyte. Microscopic investigation of vital organs/tissues from mice given with BN@RBCM only showed the tissue appeared normal. There were no remarkable morphological changes (score 0) and no pathological lesions or inflammatory infiltration in the histological sections of the organs in these samples from the treatment groups under light microscopy (400×) (Figure 7). All these results indicated that there was little apparent subacute toxicity of BN@RBCM at doses below 134.65 mg/kg.

## 4. Discussion

BN nanofragments tended to be unstable and further spontaneously aggregated to form BNNPs. Firstly, to improve the physiological stability and dispersibility, BN@RBCM was also modified with the DSPE-PEG2000-COOH. The polydispersity index of BN@RBCM was less than 0.3, indicating the narrow particle size distribution. All the above results indicated that the BN@RBCM exhibited excellent dispersibility and stability, which was attributed to the successful coating of the RBC membrane. Compared with other boron-containing drugs, erythrocyte membrane-coated boron nitride nanoparticles have greater potential to be applied in boron neutron capture therapy. Secondly, we injected a range of doses of BN@RBCM into mice and observed and recorded acute toxic reactions. The results showed that at higher concentrations, once toxic reactions occurred, the time of death of the mice was very early and the surviving mice could quickly return to normal state without any side effects. We analyzed that this toxic reaction was probably caused by the fact that BN@RBCM was more likely to aggregate at higher concentrations. Then, we continued to experiment with the subacute toxicity of BN@RBCM in mice by repeated administration. The results, which showed toxic signs from survival mice at a dose of lower than 490 mg/kg in the acute toxicity test and all mice in the subacute toxicity test, indicated no obvious abnormality between the treated groups and the control groups. All the obtained results confirmed the good in vivo biocompatibility of BN@RBCM at the appropriate injection dose.

## 5. Conclusions

The BN, a kind of novel potential biomaterial, is a poorly water-soluble nanoparticle, so we fabricated highly soluble BN@RBCM. In this study, we evaluate the acute and subacute toxicity of BN@RBCM by intravenous injection. We determined that the approximate 50% lethal dose (LD50) of BN@RBCM was 258.94 mg/kg by the acute toxicity study. No significant toxic reaction was observed when the single dose of BN@RBCM was lower than 168.07 mg/kg. In subacute toxicity, the mice were repeatedly administered 26.93, 67.33, and 134.65 mg/kg of BN@RBCM, respectively, once every other day for 14 days. There were no obvious anomalies in body weight, organ coefficient, hematology, blood biochemistry indexes, and histopathology of the mice treated with BN@RBCM. Our study results demonstrated the biocompatibility of BN@RBCM and provide a basis for their further biomedical applications in the future.

## Figures and Tables

**Figure 1 jfb-14-00181-f001:**
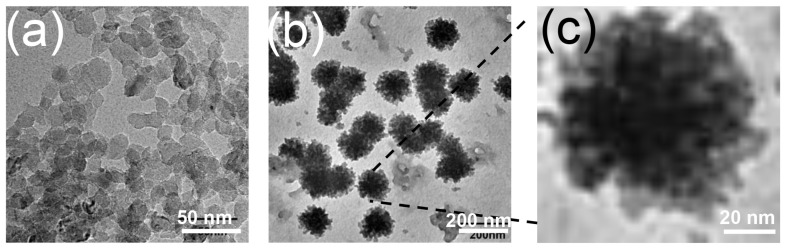
Transmission electron microscopy images of bare BNNPs (**a**) and BN@RBCM (**b**). (**c**) Magnification transmission electron microscopy image of selected nanoparticles in (**b**).

**Figure 2 jfb-14-00181-f002:**
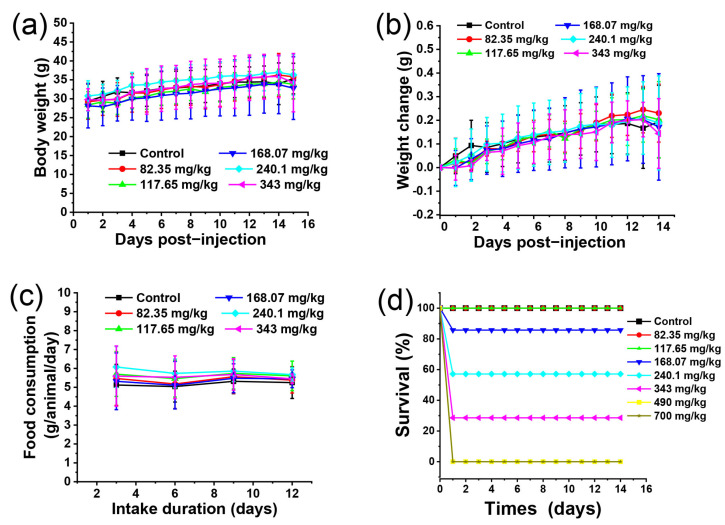
Body weight (**a**) and weight changes compared with 0 days (**b**) of mice by intravenous injection of various doses of BN@RBCM. (**c**) All groups had the same trend in food consumption per animal/day after BN@RBCM treatment once. It showed all groups had the same trend in food consumption. (**d**) The survival state of mice after BN@RBCM treatment once. Data are shown as mean ± SD, *n* = 5.

**Figure 3 jfb-14-00181-f003:**
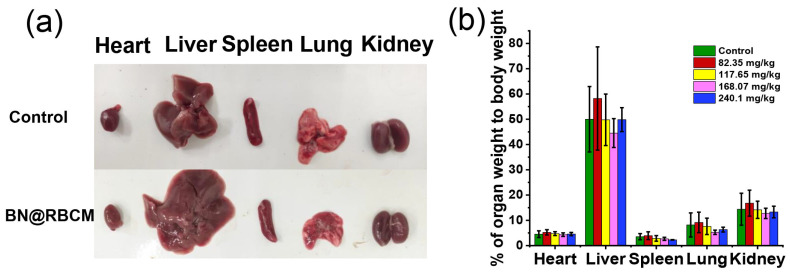
(**a**) Photograph of organs/tissue of mice received saline and 117.65 mg/kg BN@RBCM. (**b**) Organ weight to body weight ratio for mice that were administrated at indicated doses. Data are shown as mean ± SD, *n* = 5.

**Figure 4 jfb-14-00181-f004:**
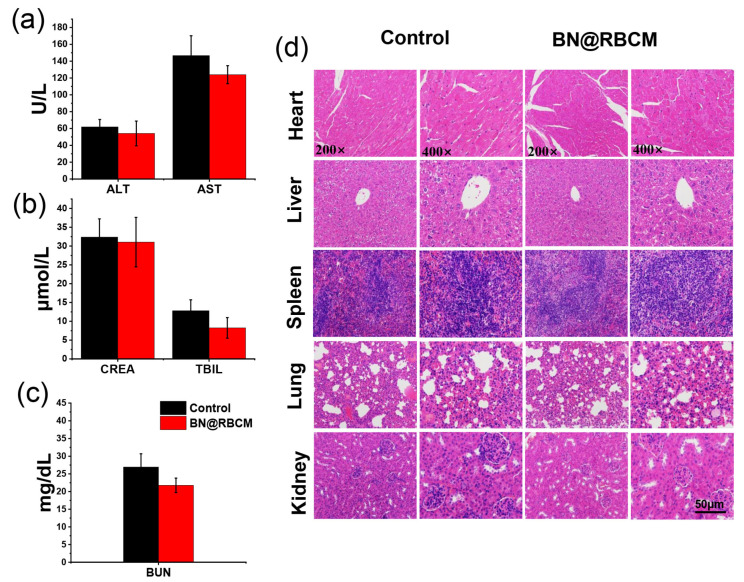
(**a**–**c**) Blood biochemistry parameters including liver-function markers (ALT, AST, TBIL) and kidney-function markers (BUN, CREA). (**d**) H&E-stained slice images of major organs of the experimental groups and control group, 14 days after single-dose administration of BN@RBCM (117.65 mg/kg) in acute toxicity study (200–400×). The data represent the mean ± SD (*n* = 5).

**Figure 5 jfb-14-00181-f005:**
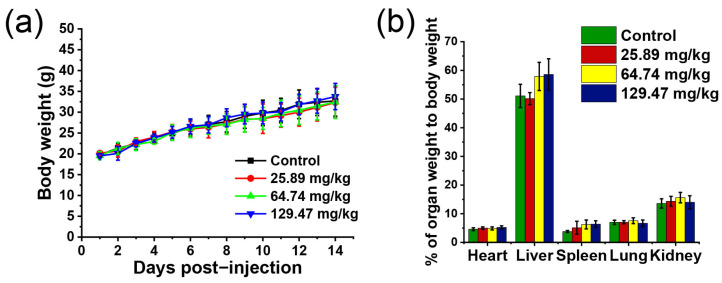
In vivo subacute toxicity evaluation in injection of the indicated concentration of BN@RBCM. (**a**) Mice body-weight-change. (**b**) The ratio of organ weight to body weight. Data are mean ± SD.

**Figure 6 jfb-14-00181-f006:**
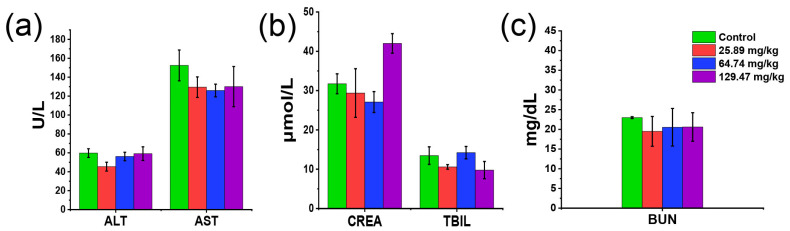
(**a**–**c**) Blood biochemistry indexes of subacute toxicity evaluation. Aspartate aminotransferase (AST), alanine transaminase (ALT), total bilirubin (TBIL), creatinine (CRE), and blood urea nitrogen (BUN). The data represent the mean ± SD (*n* = 5).

**Figure 7 jfb-14-00181-f007:**
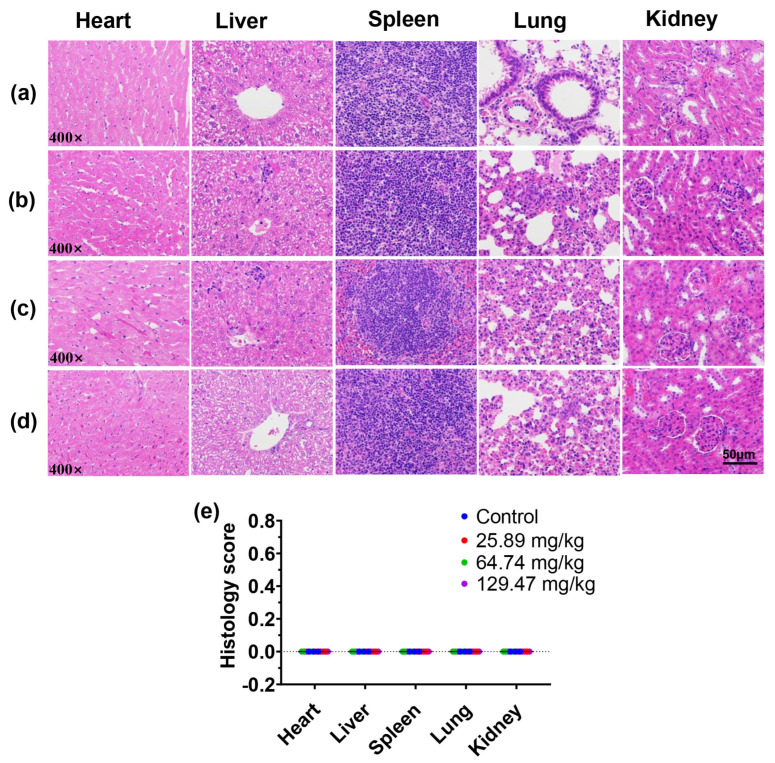
Histopathological images of the indicated organs in mice administered with saline and BN@RBCM: (**a**) 0, (**b**) 25.89, (**c**) 64.74, and (**d**) 129.47 mg/kg (scale = 50 μm). It showed no abnormality of morphological or structural. (**e**) Histology score of mice tissue slices in different treatment groups.

**Table 1 jfb-14-00181-t001:** The size distribution of BN and BN@RBCM used in this study (mean ± SD, *n* = 3).

Groups	Mean Diameter (nm)	Polydispersity Index
BNNPs	125.81 ± 0.86	0.31 ± 0.01
BN@RBCM	137.71 ± 3.36	0.29 ± 0.04

**Table 2 jfb-14-00181-t002:** Mortality of animals post intravenous injection of various BN@RBCM (*n* = 7).

Formulation	Dose (mg/kg)	Animals (N)	Death (n)	Mortality (%)	LD50 (mg/kg)
	82.35	7	0	0	
	117.65	7	0	0	
	168.07	7	1	14.3	
BN@RBCM	240.1	7	3	42.9	258.94
	343	7	5	71.4	
	490	7	7	100	
	700	7	7	100	

PS: 95% confidence interval is 212.0313–316.2278 mg/kg.

**Table 3 jfb-14-00181-t003:** Effect of BN@RBCM on the relative organ weights of mice in post-acute toxicity study.

Group	Dose (mg/kg)	*n*	Organ Weight to Body Weight Ratio				
			Heart	Liver	Spleen	Lung	Kidney
Control	(-)	6	4.511 ± 1.34	49.995 ± 12.94	3.529 ± 1.20	8.171 ± 4.76	14.388 ± 6.33
BN@RBCM	82.35	7	5.238 ± 1.05	58.207 ± 20.43	3.915 ± 1.53	9.155 ± 4.06	16.781 ± 5.15
	117.65	7	4.737 ± 0.76	49.773 ± 10.19	2.837 ± 1.11	7.602 ± 3.24	14.154 ± 3.40
	168.07	6	4.328 ± 0.70	44.533 ± 5.71	2.666 ± 0.61	5.262 ± 0.86	12.740 ± 2.03
	240.1	4	4.563 ± 0.65	49.852 ± 4.73	2.272 ± 0.11	6.332 ± 0.95	13.296 ± 2.29

PS: Data are presented as mean ± SD.

**Table 4 jfb-14-00181-t004:** Effect of BN@RBCM on hematological parameters of mice that received a single dose.

Hematological Parameters	Control Group	BN@RBCM	Normal Range
WBC (×10^9^/L)	4.1 ± 1.73	3.54 ± 2.07	0.8–6.8
LYM (×10^9^/L)	2.97 ± 1.17	2.82 ± 1.64	0.7–5.7
Mon# (×10^9^/L)	0.17 ± 0.06	0.06 ± 0.05	0.0–0.3
Gran# (×10^9^/L)	0.97 ± 0.51	0.66 ± 0.42	0.1–1.8
Lymph (%)	73.57 ± 4.79	78.94 ± 3.35	55.8–90.6
Mon (%)	3.90 ± 0.17	2.96 ± 0.18	1.8–6.0
Gran (%)	22.53 ± 4.62	18.10 ± 3.18	8.6–38.9
RBC (×10^12^/L)	8.77 ± 1.51	8.97 ± 0.54	6.36–9.42
HGB (g/L)	136.00 ± 19.92	130.20 ± 8.20	110–143
HCT (%)	42.97 ± 8.96	44.02 ± 1.80	34.6–44.6
MCV (fL)	48.87 ± 1.99	49.22 ± 2.20	48.2–58.3
MCH (pg)	15.53 ± 1.17	14.46 ± 0.23	15.8–19
MCHC(g/L)	319.00 ± 26.29	295.20 ± 9.98	302–353
RDW (%)	16.27 ± 0.51	15.70 ± 0.43	13–17
PLT (×10^9^/L)	674.67 ± 444.02	688.00 ± 230.93	450–1590
MPV (fL)	6.40 ± 0.69	5.50 ± 0.51	3.8–6.0
PDW	16.97 ± 0.57	16.72 ± 0.40	15–17
PCT (%)	0.41 ± 0.24	0.37 ± 0.10	0–0.5

PS: Data are presented as mean ± standard deviation. *n* = 5.

**Table 5 jfb-14-00181-t005:** Effect of BN@RBCM on the ratio of organ weight to body weights of mice in subacute toxicity study of 14-day repeated administration.

	Organ	Control	Dose (mg/kg)		
			25.89	64.74	129.47
♀ female	Heart	4.559 ± 0.004	4.887 ± 0.41	5.316 ± 0.50	4.854 ± 0.17
	Liver	48.830 ± 2.85	50.906 ± 1.64	62.788 ± 2.16	54.847 ± 1.25
	Spleen	3.907 ± 0.34	5.749 ± 2.91	7.418 ± 1.09	6.964 ± 0.05
	Lung	7.749 ± 0.37	7.419 ± 0.16	8.596 ± 0.67	5.774 ± 1.52
	Kidney	15.326 ± 1.74	13.632 ± 0.35	14.732 ± 0.93	11.994 ± 1.25
♂ male	Heart	4.644 ± 0.70	5.153 ± 0.44	4.582 ± 0.35	5.440 ± 0.65
	Liver	52.619 ± 4.51	49.03 ± 2.56	54.622 ± 2.45	61.065 ± 5.96
	Spleen	3.715 ± 0.41	4.117 ± 1.61	5.480 ± 1.41	5.970 ± 1.48
	Lung	6.495 ± 0.34	6.431 ± 0.10	6.866 ± 0.10	7.271 ± 0.42
	Kidney	14.426 ± 1.52	15.356 ± 1.18	16.220 ± 2.21	15.326 ± 1.74

PS: Data are presented as mean ± SD, *n* = 5.

**Table 6 jfb-14-00181-t006:** Effect of repeated dose of BN@RBCM on hematological parameters of mice.

Hematological Parameters	Control Group	Normal Range	BN@RBCM
WBC (×10^9^/L)	3.97 ± 0.74	0.8–6.8	5.24 ± 1.96
LYM (×10^9^/L)	3.12 ± 0.63	0.7–5.7	4.18 ± 1.89
Mon# (×10^9^/L)	0.12 ± 0.04	0.0–0.3	0.16 ± 0.09
Gran# (×10^9^/L)	0.73 ± 0.19	0.1–1.8	0.90 ± 0.20
Lymph (%)	78.50 ± 4.12	55.8–90.6	77.82 ± 6.08
Mon (%)	3.22 ± 0.87	1.8–6.0	3.58 ± 0.73
Gran (%)	18.28 ± 3.28	8.6–38.9	18.60 ± 5.58
RBC (×10^12^/L)	8.92 ± 0.26	6.36–9.42	9.26 ± 0.58
HGB (g/L)	130.83 ± 3.71	110–143	132.60 ± 7.80
HCT (%)	49.42 ± 1.52	34.6–44.6	51.04 ± 3.28
MCV (fL)	55.45 ± 1.58	48.2–58.3	55.16 ± 1.33
MCH (pg)	14.60 ± 0.43	15.8–19	14.28 ± 0.28
MCHC(g/L)	264.33 ± 6.06	302–353	259.20 ± 3.27
RDW (%)	16.38 ± 0.69	13–17	16.80 ± 0.58
PLT (×10^9^/L)	1098.83 ± 278.60	450–1590	1129.20 ± 483.97
MPV (fL)	5.82 ± 0.56	3.8–6.0	5.10 ± 0.37
PDW	16.90 ± 0.46	15–17	16.26 ± 0.36
PCT (%)	0.64 ± 0.17	0–0.5	0.56 ± 0.21

PS: Data are presented as the mean ± SD. Number of animals (*n* = 5).

## Data Availability

The data that support the findings of this study are available from the corresponding author upon reasonable request.

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
