# Peer review of "Acute and Subacute Toxicity Evaluation of Erythrocyte Membrane-Coated Boron Nitride Nanoparticles"

_jfb, 2023, doi:10.3390/jfb14040181_

Round 1
Reviewer 1 Report
Overview:
The authors prepared RBC-enveloped boron nitride nanoparticles and assessed their stability and potential in vivo toxicity in mice. They used the results of histopathological studies of major organs, hematology, blood chemistry together, with physical changes in the mice and LD50 values to conclude that the BN nps possessed low toxicity and good biocompatibility. The manuscript is well-written, and the study methods and conclusions are sound, but major revisions are needed before this manuscript can be published.
Comments:
1. This study is similar to another published paper on “RBC membrane camouflaged boron nitride nanospheres for enhanced biocompatible performance” https://doi.org/10.1016/j.colsurfb.2020.110964. The authors should explain how their work differs from the published paper and clearly state the novelty of their work in the introduction.
2. Lines 51-52: The sentence “Most of the collected data…” needs to be revised to make it clearer and easy to understand.
3. Line 72: Check the chemical formula for boron nitride. Is it H3BO3?
4. Line 101: “…washing with PBS in Millipore.” Did you mean Millipore water?
5. Line 105: “All animals were treated accordance…” should be “…treated in accordance…”
6. Animal experiments: Why was 14-days chosen for the study?
7. Results: DLS typically measures the hydrodynamic size of particles and not the true size. The authors should make this clear in their write-up. The authors may also estimate the average true size from image analysis of the TEM micrographs.
8. Results: The authors should explain how they obtained the LD50 value.
9. Lines 189- 190: “…and the organ coefficient…Fig(3b).” Could you make this statement clearer and easy to understand?
10. Line 203: “… shown in indicated…” Remove “in”
11. Results: The authors showed the results (hematology, blood chemistry, histopathology) for the lowest dose (117.65 mg/kg). Could they comment on the effect of the higher dose? Are the observations similar?
12. Lines 236-237: “The … were within the normal range.” There are two data points (HCT and MCHC) that are outside the normal range. So, I suggest “… were mostly within the normal range.”
13. Discussion: The authors should also the implications of their findings. For example, how do their nanoparticles compare with other nps in the literature designed for similar applications?
14. What advantage does RBC membrane coating offer to the structure/stability compared to other cell membrane coating? Why was RBC membrane coating chosen?
Reviewer 2 Report
The research reported in this article is neatly planned and performed. The synthesis part is usual, but executed well, the procedures reported and characterization techniques used are adequate, the data is acceptable and comparable with the literature. The biological studies are thorough and detailed. This article will be a good addition to the literature.
English used in the article is clear and understandable. The narrative of the article is also direct and reader friendly. I recommend this article for the publication in the current form.
Reviewer 4 Report
1. Summary of strengths, weaknesses, and overall contribution:
The authors reported a comprehensive toxicity study of the erythrocyte membrane-coated boron nitride nanoparticles in mice. The physical properties of the membrane-coated nanoparticles were characterized, and both acute and subacute toxicities of the nanoparticles were studied on different levels. This study provides insights into the safety of boron nitride-based nanoparticles and is useful for their development and clinical use. Overall, the manuscript can be considered for publication after some minor revisions are made to further improve the quality of the manuscript.
2. Major comments:
1) It would be meaningful to know what effects the membrane coating may have on the toxicity of the BNNPs. Therefore, the toxicity of BNNPs can also be studied as a control for the comparison with BN@RBCM, at least in an acute lethal toxicity study. Otherwise, the authors may comment on the impact of membrane coating on the in vivo toxicity of such materials.
2) In Figure 6b, for the 129.47 mg/kg group, the CREA looks significantly higher than the control. Is that true? If so, that means the high dose may have toxicity to the kidney. The lethal effects may result from the dysfunction of the kidney, which can also be investigated. The authors need to further explain this data.
3. Minor comments:
1) For the characterization of the nanoparticles (Figure 1), what is the difference between bare BN and BNNPs? Could the authors also show the TEM images of BNNPs to reveal the possible morphology difference among BNNPs, bare BN, and BN@RBCM?
Round 2
Reviewer 1 Report
The authors have made a lot of effort to revise the manuscript to improve its quality. However, a few minor revisions are still needed:
1. The reviewer is of the view that the authors cite this paper https://doi.org/10.1016/j.colsurfb.2020.110964 in their manuscript as it is an important study with similarities to the current work.
2. The reviewer made this comment in the first review:
“Results: The authors showed the results (hematology, blood chemistry, histopathology) for the lowest dose (117.65 mg/kg). Could they comment on the effect of the higher dose? Are the observations similar?”
The authors responded cogently to this comment, but it wasn’t captured in the manuscript. Could they incorporate their response in the revised manuscript?
